# Roles of β-Endorphin in Stress, Behavior, Neuroinflammation, and Brain Energy Metabolism

**DOI:** 10.3390/ijms22010338

**Published:** 2020-12-30

**Authors:** Alexander Pilozzi, Caitlin Carro, Xudong Huang

**Affiliations:** Neurochemistry Laboratory, Department of Psychiatry, Massachusetts General Hospital and Harvard Medical School, Charlestown, MA 02129, USA; apilozzi@mgh.harvard.edu (A.P.); ccarro@bu.edu (C.C.)

**Keywords:** β-endorphins, behavior, brain energy metabolism, neurodegeneration, neuroinflammation, psychiatric disorders, stress

## Abstract

β-Endorphins are peptides that exert a wide variety of effects throughout the body. Produced through the cleavage pro-opiomelanocortin (POMC), β-endorphins are the primarily agonist of mu opioid receptors, which can be found throughout the body, brain, and cells of the immune system that regulate a diverse set of systems. As an agonist of the body’s opioid receptors, β-endorphins are most noted for their potent analgesic effects, but they also have their involvement in reward-centric and homeostasis-restoring behaviors, among other effects. These effects have implicated the peptide in psychiatric and neurodegenerative disorders, making it a research target of interest. This review briefly summarizes the basics of endorphin function, goes over the behaviors and regulatory pathways it governs, and examines the variability of β-endorphin levels observed between normal and disease/disorder affected individuals.

## 1. Introduction

β-Endorphins are peptides that exert effects throughout the body. In the brain, they are considered to be both neurotransmitters and neuromodulators, as they have the ability to elicit more stable and long-lasting effects on more distant targets than typical neurotransmitters [1]; β-endorphins exhibit a notably high degree of degradation-resistance in the brain [1]. β-Endorphins are produced primarily by both the anterior lobe of the pituitary gland [2,3], and in pro-opiomelanocortin (POMC) cells primarily located in the hypothalamus [3,4]. β-Endorphin and other cleavage products are produced in the multistage processing of POMC primarily involving prohormone convertases (PC) 1 and 2. PC-1 cleaves POMC into adrenocorticotrophic hormone biosynthetic intermediate and β-lipotropic hormone. PC-2 cleaves β-lipotropic hormone into β-endorphin and γ-lipotropic hormone [5]. Carboxypeptidase-E (CPE) is also involved in the processing of POMC and with the removal of C-terminal basic residues (arginine/lysine) which are left after PC cleavage. Notably, however, mice with inactive CPE (CPE^fat^/CPE^fat^) produce greater quantities of β-endorphin(1–31) than wild-type mice, indicating that CPE is not required for its synthesis [6]. Additional processing, such as the acetylation of some β-endorphins, and shortening, may also occur [7,8,9].

In the brain, the peptide and other related proteins are most prevalent in the hypothalamus, thalamus–midbrain, amygdala, hippocampus, and brainstem [1]. Though the primary source of peripheral β-endorphin is the pituitary gland [3], β-endorphins, POMC, and PCs 1 and 2 have been identified in the skin [10,11,12], as well as cells of the immune system [13], though transcripts of POMC are notably found only at very low levels in the latter [14]. Though there is evidence that peptides such as β-endorphins can penetrate the blood–brain barrier to a degree, based on studies of radiolabeled, intracarotidally injected peptides [15], and P-glycoprotein notably is involved in the efflux of β-endorphin from the brain [16], peripheral and central CSF levels of the peptide are not necessarily related [3].

There are multiple forms of endorphins, though β-endorphin(1–31) is the only form with a potent analgesic effect. The other, more inactive forms are shorter, such as β-endorphin(1–27); notably, β-endorphin(1–31) is the primary form found in the anterior pituitary gland and brain regions such as the hypothalamus, midbrain, and amygdala [17,18], while the shorter forms are more common in the intermediate pituitary and brain regions, such as the hippocampus, colliculae, and brain stem [17,18]. Typical antibodies against β-endorphin will recognize 1–31, 1–26, 1–27, and the acetylated forms [18]. β-Endorphin(1–27) has notably reported to be an antagonist of β-endorphin(1–31) in in vivo animal studies [19,20,21], though this has been refuted in more recent in vivo studies [22] and in vitro [8,23], indicating that the shorter forms of β-endorphin are full agonists, and the studies indicating antagonistic activity were somewhat flawed [8]. There is evidence that receptors with opiate activity that preferentially bind β-endorphin(1–31) are present [1].

β-Endorphins are part of the system of opioid receptor agonists. The endorphin family includes β-endorphin, α-neoendorphin, enkephalins, and dynorphins [24]. β-Endorphins exert an analgesic effect that is more potent than morphine [1,2], and act primarily on the mu family of opioid receptors [25], which are, like the two other opioid receptors, delta and kappa, G-protein coupled receptors [24]. Naloxone, a typical antagonist of other opiates, has been found to reduce β-endorphin binding as well, and is commonly used in opioid-related studies [1]. β-Endorphins, along with other opioids, appear to attenuate cyclic adenosine monophosphate levels, and decrease calcium uptake [2]. The peptide is typically released to the periphery in response to a painful or stressful event, where they inhibit somatosensory fibers, with a focus on nociceptors [2]. It should be noted that, while β-endorphin has the highest affinity for the mu receptor, it also acts on other opioid receptors [24], particularly the delta opioid receptor [26]. Other opioid receptor agonists, such as the enkephalins and endomorphins, can also activate the mu-receptors [24,26,27,28], so all mu-receptor activity cannot necessarily be attributed to β-endorphin. It should also be noted that endomorphins are peptides which have been isolated in brain and immune tissue that exhibit the highest affinity for the mu opioid receptor [29,30,31,32], but as no precursor has yet been identified, it is unclear if they are truly endogenous [33].

β-Endorphins are related to the hypothalamic–pituitary–adrenal (HPA) axis. The HPA axis, which governs a wide range of functions, including metabolic and immune responses [34,35], is heavily involved in the body’s stress-response, with stress being defined as the perception of a threat, real or imagined, to one’s well-being or homeostatic state [36]. The HPA is first stimulated when a stressful event induces production of corticotropin-releasing hormone (CRH). This results in the simultaneous release of Adrenocorticotropic hormone (ACTH) and β-endorphins, both of which are produced from the cleavage of POMC, and are stored and released together in secretory vesicles [37]. As the initial cleavage generates a β-endorphin precursor, β-lipotropic hormone (LPH), which requires further processing to form β-endorphin, the two are not necessarily present in a 1:1 ratio [4]. The effects of the HPA can be tied to a variety of behaviors, including exercise, drug-use, sexual behaviors, among others [38].

### Demographics of β-Endorphin Levels

How the presence and quantities of β-endorphins change as one ages is somewhat inconclusive, though animal studies primarily suggest a decrease with age. Studies involving the global and brain-region-specific levels of β-endorphins have found a consistent decline in β-endorphin levels [39,40]. Kowalski et al. found similar decreases, though they were found to be dependent on brain region; β-endorphin levels in the hypothalamus of the rats were found to decrease significantly, but there were no significant changes in other brain regions [41]. A 2010 study observing how horses of different ages responded to exercise found that post-exercise levels of β-endorphins were higher in young horses than older horses, paralleling the idea that levels of the peptide decrease overall with age [42]. Contrary to other body regions, Sacerdote et al. observed the concentration of β-endorphin in blood mononuclear cells, and found that levels increase with age in rats, suggesting a possible role in age-related immunodepression [43].

Studies involving human subjects are much less consistent than their animal counterparts. A 1992 study measuring β-endorphin levels in blood mononuclear cells found that, in normal individuals, levels of β-endorphin appeared to increase significantly after the age of thirty, then remain relatively stable up to the age of 99 [44]. This is fairly consistent with Sacerdote et al.’s findings, which also noted increased levels of β-endorphin in human mononuclear cells [43]. A 1993 study working with patients with major depression found that there was a significant negative correlation between age and daytime plasma levels of β-endorphin in controls, a trend that was not observed in the depressed group [45]. This conflicts with a 1988 study, which found no difference in plasma β-endorphin measures taken during the morning hours in young compared to old adult males [46]. Individuals experiencing lumbar or cervical disk hernias also exhibited no age-related difference in β-endorphins in CSF [47]. Overall, it would seem that older individuals have lower levels of β-endorphins, but this should not be assumed in all cases and further verification in human subjects is required. Though the mechanisms for an observed decrease is not clear based solely on observed levels of β-endorphin, expression of its precursor POMC appears to decrease with age in rats in certain brain regions [48,49], while conversely POMC expression may be increased in human skin [50].

Studies of the differences between the sexes in terms of β-endorphin levels appear to be more consistent, though significance varies. Goldfarb et al. sampled twelve healthy men and twelve healthy women and compared circulating β-endorphin concentrations both at rest and following exercise. They found that women experienced lower levels of BE both at rest, and following low-intensity exercise, though these results did not achieve statistical significance [51]. A 2006 study observing the effects of regular alcohol consumption on β-endorphin levels found that females had significantly lower levels of β-endorphins in plasma regardless of age and level of alcohol consumption (non-drinkers to alcoholics) [52]. Interestingly, this pattern appears to persist, to some degree, in infants, with premature male infants having higher concentrations of plasma β-endorphin than females, though the significance of this varied based on condition and stage of growth [53]. Studies involving rats corroborate these results, showing that that levels of β-endorphins, and responses to them, were lower in females than in males; however, these results were highly dependent on the brain-region/tissue sampled, and the type of stimulus tested [54,55,56].

## 2. Effects of β-Endorphins

### 2.1. β-Endorphins Analgesia

β-Endorphins are part of the endogenous opioid system. Like their exogenous counterparts, much of their effect is centered around relieving pain. β-Endorphin in particular have been found to be around 18 to 33 times more potent than morphine on a per-molar basis [57]. However, this effect is largely exclusive to β-endorphin(1–31), which is orders of magnitude more potent than the shorter β-endorphin(1–27) [1]. As with the exogenous opioids, the analgesic effects of β-endorphin can be reversed through the administration of naloxone [57]. Indeed, administration of an exogenous opioid for pain relief has been found to be accompanied by a corresponding decrease in β-endorphin levels in cancer patients [58], and administration of morphine has been found to disrupt the normal β-endorphin-pain response in animal models; this is thought to be why the analgesic effects of opioids are variable over the course of a day [59]. However, resting levels of β-endorphin have not been found to be a significant predictor of exogenous opioid-analgesic response [60]. The peptide plays a considerable role in managing painful events; administration of the β-endorphin inhibitor dexamethasone significantly increases levels of post-operative pain relative to a placebo [61]. Women with lower levels of β-endorphin towards the end of their pregnancy require pain-relieving medication beyond nitrous oxide more frequently than those with higher levels of the peptide [62]. Additionally, levels of β-endorphin have been found to be correlated with the maximum pain scores of patients of temporomandibular joint surgery [63], and women undergoing labor [64]. Notably, immunoreactivity of β-endorphin was found to increase in the arcuate nucleus of rats induced with a brachial plexus injury, which is associated with severe neuropathic pain [65].

Many pain-relieving methods and techniques can be attributed, at least in part, to β-endorphin fluctuations. The pain relief and “feeling of warmth” that is provided by connective-tissue massages, for example, has been associated with increased plasma levels of β-endorphin [66]. However, no significant change in serum β-endorphin was found in relation to complete back massages [67], and Morhenn et al. noted a decrease in blood levels of β-endorphins following a moderate-pressure back massage [68]. Numerous studies affirm that the effect of the common non-pharmaceutical, non-surgical pain relief method, acupuncture, is related to β-endorphins. Various acupuncture techniques, such as traditional (puncture), electroacupuncture and laser acupuncture have been found to reduce pain and induce an increase in levels of circulating β-endorphin in those experiencing chronic pain [69,70,71,72]. Somewhat conflictingly, a 2007 study examining the application of acupuncture to patients suffering from chronic lower back pain found that, under low-stress conditions, there was no significant increase in plasma β-endorphin immunoreactivity, and levels were “minimal at all times and in all treatment conditions” [73]. The authors note that β-endorphin is typically released in response to stress, rather than a painful stimulus itself [73].

Exercise, an activity found to increase levels of circulating β-endorphins, depending on type and intensity [74], has also been suggested to relieve pain in a variety of different conditions, though magnitude and significance of the effect vary [75]. Specific exercise routines designed for those with lower back problems, such as lumbar core stabilization exercise, has been found to decrease pain and increase levels of plasma β-endorphins, suggesting that the peptide is associated with the exercise-induced analgesia [76]. The analgesic nature of β-endorphins is clear, and it seems they play a significant role in non-surgical and non-pharmaceutical methods of pain relief.

### 2.2. β-Endorphins Effect on the Immune System

Immune responses, such as inflammation, are notably influenced by the HPA [35]. This influence includes glucocorticoids, such as cortisol, which is released in response to ACTH. Following increases in certain cytokines such as interferon-α (IFN-α) and interleukin (IL-6), cortisol is released [77,78,79]. Cortisol notably regulates inflammation in human models in a bi-phasic, concentration dependent manner [80]. Overall, it would appear that β-endorphins have inhibitory effects on immune responses. Administration of β-endorphins has been found to inhibit the proliferation of splenocytes, while administration of gamma globulins targeting β-endorphin was found to increase their proliferation [81]. Similar results are found in T cells; treatment with opiate antagonists was found to cause a dramatic initial increase in T-cell proliferation. Chronic treatment with an opiate antagonist was found to ultimately inhibit T-cell proliferation, as opiate receptors were eventually upregulated in response to the repeated administration of the antagonist [82]. Quantities of certain cytokines, such as IL-2 and interferon IFN-γ, are found to decrease in response to β-endorphins [83], whereas others such as IL-4, which is notably anti-inflammatory [84], are increased [83]. Results are similar from in vitro studies treating mouse splenocytes with morphine, a mu-receptor agonist. Production of IL-1β, IL-2, tumor necrosis factor (TNF) -α, and IFN-γ is dampened, while anti-inflammatory cytokine production is stimulated [85,86]. A 1999 study found that, in murine lymphoid EL-4 cell lines, exposure to β-endorphins caused an increase in the production of IL-2 [87]. IL-2 is a cytokine found to be a T-cell growth factor that is notably essential to the proliferation regulatory T cells, suggesting an overall increase in immune tolerance over provocation [88].

Despite this, some research suggests that β-endorphin’s effect on the immune system is not entirely inhibitory. Pre-incubation of spleen-derived rat lymphocyte cells with β-endorphins increases their proliferation. However, this effect was neither countered nor enhanced by naloxone, indicating that the mechanism by which β-endorphins produced the proliferative effect was not mediated by an opioid receptor [89]. Cytotoxic T lymphocytes are also found to experience increased proliferation in response to β-endorphins, though this effect can be partially blocked with opiate antagonists, indicating an interface with opioid receptors [90]. β-Endorphins have also been found to significantly increase the cytolytic activity and interferon production of natural killer cells [91,92]. Interestingly, it appears that β-endorphins both promote and inhibit production of antigen-specific antibodies in the presence of an antigen. This effect was found to not only be dose dependent, but also cell-donor dependent; the concentrations at which the promoting effect was at its highest and the concentrations at which the inhibitive effect was greatest differed by donor, and was the same for each donor cell line through multiple test periods [93].

Despite these exceptions, it seems that, holistically, β-endorphins dampen immune responses. In support of this immuno-suppressive effect, researchers administered β-endorphins or naloxone to mice, following skin grafts, and measured the time to rejection. Mice treated with β-endorphins experienced rejection later than controls, and correspondingly, naloxone treated mice rejected the transplanted tissue faster [83]. Mu-receptor agonists have been found to decrease intestinal inflammation in mice, with mu-receptor deficient mice experiencing dramatic increases in both inflammation and mortality [94]. In those with rheumatic diseases, which typically involve inflammation at joints, muscles or connective tissue, levels of immunoreactive β-endorphin are lower than in age-matched healthy individuals; a lack of the normal suppression provided by β-endorphins may contribute to the development and exacerbation of such disorders [95].

While these studies show that β-endorphins are, at least in part, direct regulators of immune and inflammatory processes, their role as stress-relievers may feed back into the immune response. Two meta-analyses of numerous studies looking at the relationship between stress and levels of circulating inflammatory mediators found that IL-6 and IL-1β were significantly increased following exposure to a stressor [96,97]. Marsland et al. found that levels of TNF-α had also increased significantly in response to stress, based on the studies analyzed by their team [96]. Such an increase of the body’s inflammatory responses has been implicated in a variety of disorders, both physical and psychological [98,99,100,101,102,103,104]; the attenuation of stress by opioid receptor agonists may help to keep these responses at bay. A summary of the HPA’s influence on inflammation is shown in Figure 1.

A particularly interesting exception to β-endorphin’s anti-inflammatory action takes place within the brain. In 2007, Benamar et al. found that administration of the fever-inducing agent lipopolysaccharide to the brain of mice, a fever model that is most associated with neuroinflammation, failed to induce a change in body temperature in mu-receptor knockout mice; on the other hand, normal mice exhibited a significant dose-dependent increase in body temperature in response to the injection. In contrast to its anti-inflammatory/immune effects on other regions of the body, action of the mu opioid receptor somehow induces the release of pyrogens [105]. In cases of stroke, administration of naloxone intranasally has been found to decrease microglial activation and reduce the inflammation that normally occurs [106]. Outside of the mu receptor, morphine, a common agonist of the mu receptor, has been found to interact with microglial cells, the primary immune cells of the brain, inducing them to activate via toll-like receptor 4 [107]. Morphine has unique interactions with the toll-like receptor 4/myeloid differentiation protein (MD-2) pathway of brain microglial cells. Microglial cells exposed to morphine were found to increase production of IL-1β and TNF-α, while both toll-like receptor 4 (TLR4) and myeloid differentiation factor-2 (MD-2) knockout cell lines experienced no such upregulation, implicating the receptors as integral to the neuroinflammatory response [108]. While this effect was not linked to β-endorphin, as morphine and β-endorphin primarily act on the same receptor (mu) it is not infeasible that the interaction between β-endorphin and the TLR4 and MD-2 receptors is similar to that of morphine.

### 2.3. β-Endorphins in the Holistic Stress Response

Perhaps one of the most well-studied effects of β-endorphins aside from analgesia involves a global reduction in stress-related activity throughout the body [109]. Numerous studies have been performed on rats and mice, observing the effects the presence or lack of β-endorphins has on stress-related and coping behaviors. When β-endorphin deficient mice are introduced to an aggressive, dominant mouse they were found to exhibit greater frequencies of counter-aggressive behavior in comparison to normal mice; this suggests that β-endorphins suppress this response [110]. When measuring body temperature, which fluctuates with some correlation to stress [111], they also observed that the temperature rise experienced by β-endorphin deficient mice during the stressful event was the same as normal mice, but the deficient mice returned to normal temperature faster [110].

While modulation of stress responses appears to be a relevant role of β-endorphins, it appears to be heavily involved in coping with past or continuing stressors. When researchers subjected rats to forced swimming and tail-suspension tests, which involve stressors that the subject cannot alleviate themselves, it was observed that mice with normal β-endorphin levels and activity would become immobile, making only movements necessary to keep themselves afloat/stable, sooner than β-endorphin deficient mice [112]. In order to observe the effects of β-endorphins in the aftermath of a stressful event, food deprived mice were subjected to forced swimming, then placed in a container with a pre-weighed almond slice, which represented a novel food item for the mice. β-Endorphin deficient mice were more averse to the novelty feeding, taking longer to investigate and start consuming the food; however, both groups ultimately consumed roughly the same quantity of food [113]. Interestingly, expression of the mu-opioid receptor in the hypothalamus, on which β-endorphins act, was found to decrease in response to repeated/prolonged restraint-stress in rats [114]. Rainbow trout exposed to prolonged stress via crowding for 72 h were found to maintain increased plasma levels of cortisol; keeping cortisol levels high via continuous administration has been found to attenuate plasma concentrations of β-endorphins.

β-Endorphins are similarly involved in human stress responses. Observing levels of β-endorphins found in serum samples of surgical patients finds that there is a significant rise in serum levels of the peptide resulting from surgical stress [115]. Interestingly, a similar spike in pre-op and post-op serum levels of β-endorphins have been observed in preschool age surgical patients, with a more moderate spike in neonates and no significant change in infants; the notable difference in preschool age patients may be attributed to their increased emotional perceptivity compared to younger age groups [116]. Looking at more benign sources of stress, we find that high-stress activities, such as skydiving, can induce transient spikes in blood β-endorphin levels [117,118], and that this response is lessened in those that are less anxious prior to engaging [117]. However, more prolonged sources of stress in humans, such as academic stress, cortisol levels were found to increase and stay elevated leading up to examinations, while β-endorphin levels did not change significantly during the same period [119]. These studies lend credence to the idea that β-endorphins are primarily involved in attenuating acute stress responses, and not necessarily in the handling of prolonged (or chronic) stressors.

### 2.4. β-Endorphins and Behavior

Much of β-endorphin’s governance of behavior has to do with either brain reward-system pathways, among other changes in food-consumption, sexual behaviors, among others [109]. Though much focus on reward systems, and how it relates to many of the same behavioral changes as β-endorphin, is given to dopamine [120,121,122,123,124,125], β-endorphins are also active in various reward-system pathways. In fact, β-endorphins appear to exert regulatory effects on serotonin, inhibiting its release and modulating its turnover in a region-dependent manner that can be nullified with opioid antagonists [126,127,128]. Reciprocally, serotonin appears to regulate the secretion of β-endorphins in a similarly region-dependent way. Notably, secretion is increased in the hypothalamus and decreased in the hypothalamus in response to serotonin [129,130,131]. 

While not necessarily independent of other factors, the peptide is heavily involved in certain reward pathways. It is known that β-endorphins inhibit the release of gamma-aminobutyric acid (GABA) [132], the primary inhibitory neurotransmitter of the brain [133], which can lead to excess accumulation of dopamine, a key agent associated with feelings of pleasure [132]. Indeed, binding of opiates in general to their respective receptors is associated with feelings of well-being and euphoria [134]. Rats given access to levers that administer endorphins directly to their brain will self-administer the peptide to themselves regularly [135]. Injections of endorphins and exogenous opioids such as heroin can produce conditioned place preference, a common test for drug-reward [136], in rats, suggesting opioids’ involvement in reinforcement and reward [135,137]. Consumption of highly palatable foods in non-food-deprived rats has been observed to increase binding of β-endorphin in the hypothalamus, further reinforcing β-endorphin’s role in reward pathways [138].

#### 2.4.1. Addiction

Much of the research on β-endorphin’s human effects has to do with addictions, particularly those involving drug use and alcohol abuse. It is integrally involved in producing the feelings of euphoria associated with certain drugs and has been implicated in the development of alcoholism [139]. Consumption of cocaine, for example, leads to an increase in plasma concentrations of β-endorphins [140]. Opioid antagonists have been found to alleviate cocaine-addiction behaviors, as cocaine-induced reinforcement of conditioned place preference (CPP) was reduced through the administration of naltrexone, and reinforced by methadone [141]. Interestingly, the rewarding action of cocaine does not appear to be mediated by the mu-opioid receptor; mu-receptor knockout mice exhibit cocaine induced CPP, but β-endorphin deficient mice do not [142]. β-Endorphin has been suggested to interact with the delta opioid receptor in cocaine addiction, which may not only mediate the direct reward, but also the incubation of craving that can induce relapses in abstinent recovering cocaine addicts [143]. Opioid drugs such as heroin directly interface with opioid receptors, inducing pain relief and stimulating reward centers [144]. Opioid antagonists, in particular naloxone, are frequently used in the treatment of heroin addiction, through tandem treatment with a low-risk opioid such as buprenorphine [145], and heroin overdoses [146]. Certain variations of the mu-opioid receptor have been found to influence the condition and treatment outcome of heroin addicts [147]. Alcohol consumption appears to stimulate the release of β-endorphin, but habitual consumption ultimately results in a reduction in β-endorphin levels [148]. Those with genetic deficiencies in β-endorphin levels are more likely to become alcoholics [148], and excessive alcohol consumption in rats can be curbed through the use of opiate antagonists [149], further cementing the peptide’s role in the development of alcoholism. β-endorphin has also been suggested to be related to the phenomenon of exercise addiction; increases in plasma levels of the peptide is often observed in strenuous exercise, and is associated with feelings of well-being and euphoria similar to that observed in drug addiction [150,151]. However, difficulties in relating circulating β-endorphins to brain levels of the peptide makes verification difficult [150].

#### 2.4.2. Food Consumption

Many opioid receptors, which can be found throughout the brain, are localized in brain regions that pertain to food/energy homeostasis [152]. A 1979 study found that administration of the opioid antagonist naloxone suppresses eating even in food-deprived rats [153], and successive studies have gone on to confirm this, and note that opioid antagonists reduce overall food intake in mice, including those that are genetically obese [152]. Single nucleotide polymorphisms of the mu-receptor gene have been found to be correlated to BMI in humans [154]. Total knockout of the mu-receptor gene decreases motivation to eat in mice, while showing no significant impact on the hedonic processing of food intake [155]; this is somewhat contradictory to β-endorphin’s role in food reward-pathways, and could indicate an alternative reward path, or that the β-endorphin reward pathway is not mediated by the mu receptor. In support of this, while knocking out β-endorphin’s primary receptor appears to have little effect on food hedonism, rats ingesting oil were found to have significantly increased levels of β-endorphin in serum and CSF. Interestingly, overall intake of the oil was found to be lowest where β-endorphin levels were highest, suggesting an inverse relationship. Pretreatment with naloxone decreased initial affinity for the oil based on licking-tests, suggesting a possible hedonic disruption [156]. 

Though it would seem that β-endorphins promote food consumption, with disruptions limiting food intake, the longer-term implications are quite different. Mice lacking the ability to produce β-endorphin were found to weigh significantly less than their wild-type counterparts after several weeks with food provided freely, suggesting that the peptide simultaneously has appetitive and anorexic effects, limiting excessive food consumption [157]. β-Endorphins appear to have peripheral function on taste and overall gastrointestinal function, suggesting that CNS action is not the only route through which β-endorphins modulate food intake [158].

#### 2.4.3. Sexual Behavior

Exogenous opioids are known to have an overall inhibitory effect on sexual behaviors [159], but the impact of opioid receptor agonists including β-endorphin are considerably more nuanced. In both male rats and humans, administration of the opiate antagonist naloxone has been shown to induce copulatory behavior and improve sexual performance [160,161,162]. It would seem, however, that β-endorphins still have their place in the mechanisms of sexual reward. Rats with access to a preferred and non-preferred chamber would, when allowed to copulate to ejaculation in a non-preferred chamber, shift their preference toward the originally non-preferred chamber even following castration. This shift in preference was reversible with naloxone, with the extent of this reverse increasing over time [163].

Females have been found to experience similar inhibition of sexual behavior by β-endorphin. Numerous studies involving the lordosis reflex of female rats note that it is diminished following infusions of β-endorphins [164,165,166,167]. However, Torii et al. found that the inhibitory effect becomes a facilitatory one if β-endorphin administration occurs within 6 h of priming with estrogen [166]. Pfaus et al. found that the inhibitive effect of β-endorphins on lordosis is dose dependent, noting that it is likely that high-affinity mu-receptor activity inhibits lordosis, whereas low-affinity receptor activity facilitates it [167]. While it is fair to say that β-endorphin follows the trend of exogenous opioids in inhibiting sexual behavior, interactions and the multiplicity of the pathways β-endorphins influence allow for notable exceptions to the rule.

#### 2.4.4. Other Effects

Aside from these regulatory effects on behavior, β-endorphins have their hand in other actions as well. Injection of β-endorphins to the brain has been found to induce tremors and jerking movements of the head, ocular fixation on empty space, pupillary dilation, and overall excitation in cats. The duration and severity of these effects was dose dependent, with the effects of a 12.5 µg dose lasting approximately one hour [168]. Intraventricular injections have also been observed to induce wet-dog shakes in rats [169,170], with the frequency and occurrence of the shakes depending on the ambient temperature; correlation of the wet-dog shake behavior to increased body temperature suggests a possible role in thermoregulation [170]. However, levels of β-endorphins have not been found to increase in proportion to heat loss in cold water immersion tests [171].

### 2.5. β-Endorphins in Sleep/Sleep Cycles

Early research suggests that β-endorphin modifies the sleep–wakefulness cycle [172]. Electromyogram comparisons for male cats with administration of β-endorphin or morphine find inhibited sleep through all six hours of the experiments. No rapid eye movement (REM) sleep was detected and there were numerous episodes of light slow wave sleep, as morphine impacts the ventrolateral preoptic nucleus to promote sleep [173]. In children with sleep apnea, increases in CSF levels of β-endorphin were observed in comparison to controls. The apnea was also alleviated through the administration of naltrexone, suggesting that opioid receptor activation is involved in the pathogenesis of apnea [174]. Β-endorphin may prolong insomnia and induce mania behavior in sleep deprived rats, whereas pharmacological drugs may improve the condition [175,176]. REM sleep deprivation experiments on opioid peptides, specifically β-endorphin, suggests differential changes in β-endorphin concentration depending on three areas: the hypothalamus, anterior lobe of the pituitary gland, and the blood plasma [177]. REM sleep deprivation seems to increase β-endorphin levels in the blood plasma, while decreasing levels in the hypothalamus, with no significant changes in the pituitary gland concentration. Recent sleep medicine research suggests that worsened sleep quality, reduced REM sleep, and prolonged slow wave/stage one sleep may result from administration of β-endorphin to sleep disorder patients [178]. Numerous results suggest that β-endorphin may worsen the impact of chronic sleep disorders, and benzodiazepine antagonist drugs remain a promising mechanism for improving natural sleep [178]. Further research in the field may facilitate discovery of more natural remedies for sleep disorder patients.

### 2.6. β-Endorphins in Depression and Stress/Anxiety Disorders

Given the role of β-endorphins in modulating stress responses, it should come as no surprise that various examples of psychiatric disorders can be attributed, at least in part, to some aberrance in β-endorphin levels. With regards to one of the most prevalent disorders at present, depression [179], results seem to be inconclusive. A 1992 study observing plasma levels of β-endorphin and natural killer cell (NK) activity, as β-endorphins are known to enhance NK activity, found that both indicators were significantly reduced in clinically depressed patients [180]. Given β-endorphin’s role in producing feelings of euphoria and well-being [134], one might expect levels to be consistently low in depressed patients. However, another 1993 study looking at 26 patients with major depression (MAD) and 25 controls found that plasma β-endorphin levels were significantly elevated in MAD patients [45]. Similarly, significantly higher levels of β-endorphin were observed in the blood mononuclear cells of depressed patients [181]. 

These findings are somewhat consistent with the effects of an often-mentioned treatment for depression: exercise. The results of randomized controlled trials involving depression patients found that exercise was significantly more effective than non-treatment, and provided a more modest effect when combined with other treatments, such as antidepressants or therapy [182]. While effects vary by type and duration, exercise has an immediate effect of increasing levels of circulating β-endorphins [74]. However, analysis of the basal-levels of β-endorphins at rest in middle-aged men who jogged regularly, and sedentary individuals of similar age found that they were lower in the jogging population. The jogging population also experience lower levels of depression per the MMPI scale [183]. This indicates that the long-term effects of regular exercise may be a reduction in β-endorphin levels rather than an increase. It should be noted, however, that the sample size for this study was relatively small [183].

It should also be noted, however, that depression is not a monolithic disease; there is a high degree of variation in diagnosis, treatment response and the overall course of the disease between patients [184]. These inconsistencies, and the relatively small sample sizes of most studies, may be resulting in the variable results and the presence/lack of significance in their findings. It also warrants mention that multiple works have found little to no correlation between levels of β-endorphin in blood plasma, and β-endorphin levels in cerebrospinal fluid (CSF) [185,186,187]. Exceptions to this can arise in conditions involving disruption of the blood–brain barrier, such as cerebral malaria [188]. However, studies involving CSF concentrations of the peptide in patients with depression and chronic pain found no correlation between CSF β-endorphins and depression [189,190]. β-Endorphin levels were stable in those with lower back-pain even after treatment of pain and the resolution of depression [190]. Sufferers of chronic migraines exhibited significantly lower levels of CSF β-endorphin, but the effect was not exacerbated by the presence of depressive symptoms [189]. While it is apparent that both peripheral and CSF β-endorphins exert effects on mental state, they can be independent of each other, and thus should be considered separately.

β-Endorphins hold considerable sway over stress-related cognition. The peptide is most notably involved in conditions related to states of anxiety and hyperarousal, with a greater association between states of hyperarousal and fluctuations in β-endorphin levels [191]. Hyperarousal is associated with symptoms of trauma and intrusion along with an increase in anxiety, while depressive symptoms, avoidance and pain are most related to states of anxiety [191]. Endorphin release is noted when post-traumatic stress disorder (PTSD) sufferers are asked to recount their traumatic experiences, with the recollection producing an analgesic effect that is countered with opiate blockers, such as naloxone [192]. A 1989 study involving 21 PTSD patients and 20 controls found that the PTSD sufferers had significantly lower plasma concentrations of β-endorphins. Studies involving PTSD-like afflicted rats finds similar results, with lowered plasma β-endorphin concentrations before, during, and after re-exposure to a trauma cue compared to non-PTSD like rats. Assays of the removed brain-tissue of PTSD-like and non-PTSD like rats found significantly lowered levels of β-endorphin in the amygdala of PTSD-like group [193]. Overall, it would seem that β-endorphins have a great degree of involvement in stress and anxiety-related disorders [194].

Though not necessarily tied to a discrete psychiatric disorder, there are also notable abnormalities in the β-endorphin levels of those who have attempted or committed suicide. In a sample of 37 patients hospitalized for either a suicide attempt or suicidal ideation, a fairly strong, significant (*r* = 0.702, *p* = 0.007) correlation was found between the number of lifetime suicide attempts and plasma levels of β-endorphin. In addition, the suicide-attempters were found to have higher pain-thresholds than those who had only exhibited suicidal ideations; the role of β-endorphin in analgesia may be the link between these two observations [195]. It should be noted that an earlier study found no correlation between varying suicidal subgroups in a study of 44 patients who had attempted suicide, and levels of CSF β-endorphin [196]. This is consistent with the general pattern found in depression patients: high circulating β-endorphins, and normal levels of CSF β-endorphins. Comparisons of post-mortem brain samples of suicide victims and cases of sudden death revealed further abnormalities in patterns of β-endorphin levels, which were found to be asymmetrical in suicide victims, with lower concentrations of β-endorphin found in regions of the brain’s left hemisphere when compared the same regions in the right hemisphere and controls [197]. Further supporting the idea that β-endorphins may be relevant to suicide, a 2008 study found that a variant of the mu-opioid receptor, A118G, was associated with an enhanced risk of suicide [198]. Taking these observations together, it seems that β-endorphins may have some relevance to suicide, making them a potential target for reducing suicide risk.

### 2.7. β-Endorphins in Brain Health and Neurodegenerative Disorder

One of the most prevalent neurodegenerative disorders of today is Alzheimer’s disease (AD). Affecting over 5.8 million Americans, 10% of those 65 and older, a number that is expected to only grow over time [199]. While the primary characteristics of the disease involve the formation of senile plaques and neurofibrillary tangles in the brain, there is notable alteration in neuropeptide-containing neurons found in postmortem analysis of AD-afflicted brains [200]. A 1999 study on infusions of the opioid antagonist naloxone found that cortisol levels in AD patients were more elevated, with that elevation lasting longer, than in controls. The AD patients also experienced hypothermic responses to 2.0 mg/kg naloxone; overall, this data verifies an HPA axis disruption in AD, one that in part involves the opioid system [201]. Multiple studies have found that CSF levels of β-endorphins are decreased in those with Alzheimer’s dementia [185,202,203]. Levels of the peptide have been suggested to be based on the cause of dementia, but not its severity [204]. However, in persons with AD, Jolkkonen et al. found that those with more severe cases of Alzheimer’s dementia exhibit lower levels of β-endorphin than those with more moderate cases [202]. Looking outside β-endorphin itself, there is evidence for alterations in the brain’s opioid receptors found in those with AD, based on postmortem brain analysis.

Interestingly, a 2008 study looking at exercise-induced adult neurogenesis in mice found that β-endorphins are integrally involved in the process. β-Endorphin deficient mice had completely blocked the increase in cell proliferation due to exercise that was observed in wild-type mice [205]. Lower levels of β-endorphin in the brains of older individuals and AD patients may be negatively impacting neurogeneration and plasticity of the brain [206].

#### 2.7.1. Neuroinflammation 

Neurodegenerative diseases such as AD, often exhibit neuroinflammation as a prominent feature. Interestingly, the line between psychiatric and neurodegenerative disorders is somewhat blurred. Research suggests that neurodegenerative disorders are often associated with psychiatric ones. For instance, 90% of AD patients are also affected by some manner of psychiatric aberration, with depression being the most common, affecting 50% of patients [207]. Some research suggests that the link between the two disorders may have to do with neuroinflammation [208]. In both patients with AD and those with MAD, abnormal microglial activation is noted, along with a corresponding increase in inflammation as microglial cells release inflammatory cytokines [208,209]. Though the exact role of this inflammation in AD is a subject of debate, some consider it to be a “vicious cycle” of inflammatory action enhancing amyloid-β deposition, among a host of other factors, that in turn encourage inflammatory action [100,210]. As such, β-endorphins may be useful in ameliorating this inflammation, due to its direct, and indirect (stress-related) anti-inflammatory properties. However, its unique interactions with microglial cells may overwrite these benefits in the brain, instead implicating the peptide in the neuroinflammatory process. Given β-endorphin’s role in systemic and neural inflammatory responses, it warrants investigation into the peptide as a potential therapeutic target for disorders linked to neuroinflammation.

#### 2.7.2. Metabolism

Some of the other biological functions of β-endorphin may explain how abnormalities in its prevalence in the body can contribute to neurodegenerative disorder. Brain energy metabolism homeostasis is integral to its health and function and is altered in neurodegenerative disorders such as AD [211]. β-Endorphins have been observed to exhibit an indirect, but profound, role in brain energy metabolism. A mice feeding study investigating the role of hypothalamic POMC neurons and cannabinoid receptor 1 (CB1R) suggests that this receptor is localized in the mitochondria of β-endorphin producing POMC neurons [212]. Furthermore, CB1R activation during satiety showed both an increase in feeding and in β-endorphin levels. This finding suggests an indirect link between cannabinoid enhanced activation of POMC neurons to trigger neuropeptide β-endorphin release, relying on glucose brain energy storage and mitochondria. When mitochondria are blocked, there is no longer CB1R induced feeding, as POMC cells do not respond as prior to the change. Early research highlights the process of mitochondrial energy supply to the process of β-endorphin release due to initial accelerated calcium ion uptake in mitochondria through synaptosomes until capacity is reached [213]. A decline in mitochondrial activity is a notable feature of aging, and mitochondrial dysfunction is a notable feature of neurodegenerative disease. This decline appears to be correlated with reactive oxygen species (ROS) levels, indicating that oxidative damage is a likely cause [214]. 

AD in particular has been noted to be related to diabetes, with obesity and diabetes both found to independently increase one’s risk for developing AD [215]. Models of brain-diabetes often exhibit abnormalities similar to AD, and insulin deficiency and resistance are potential mediators of the neurodegenerative process, with associated disfunction of insulin receptor expression decreasing in an AD-stage dependent manner; deficiency in insulin/IGF signaling, which leads to deficiencies in energy metabolism, inflammation, and oxidative stress [216]. Notably, β-endorphins have been observed to cause an increase in plasma glucose, insulin, glucagon, and C-peptide, with this response inhibited with naloxone, in obese human subjects [217]; β-endorphin also has been found to induce hyperglycemia in rats [218].

#### 2.7.3. Growth-Factors and Neurovasculature

The opioid receptors, including the mu-receptor, have also been found to stimulate the activity of various growth factor receptors, such as the epidermal (eGFR), vascular endothelial (VEGFR), and insulin-like growth factor receptors (IGFR) [219]. Low levels of eGFR have been associated with instances of mild cognitive impairment [220], though some research indicates there is no significant association between eGFR and cognitive function [221]. VEGF is a neuronal regulator, and is integral to the development of neural dendrites and axons, as well as the neurons themselves [222]. VEGF is also an important factor in angiogenesis throughout the body and in the brain, and is notably active in neurovascular remodeling and repair in instances of damage [223,224]. The growth factor also exerts neuroprotective effects in cases of ischemia [225], and in rat models of Parkinson’s disease [226]. VEGF has been found to improve memory in AD mice, primarily through its angiogenic property [227,228]. How VEGF levels are correlated to AD is somewhat unclear, with studies giving conflicting relative concentrations of the growth factor [229,230]. Interestingly, long-term activity of the IGF-1 receptor appears to be detrimental in cases of AD, with deletion of the receptor gene in the brain promoting longer lifespans and delayed accumulation of Aβ in rats [231,232]. Despite this, some studies suggests some neuroprotective effects of IGF, protecting neurons from cell death [233], and ameliorating mitochondrial/metabolic disturbances [234]. 

Changes in neurovasculature are common features of neurodegenerative diseases such as AD and Parkinson’s disease [235]. The role of β-endorphin on the neurovascular system has been described by studies in indirect association with the blood–brain barrier. Although β-endorphin in its original form cannot be transported through the blood–brain barrier due to its large size, reduced peptides may readily cross the barrier, as β-endorphin is cleaved to intermediate tyrosine and capillary peptidase [236]. Previous studies have functionally linked nitric oxide and β-endorphin as impactful between the blood vessels and the hypothalamus. The medial basal hypothalamus of male Wistar rats, showed a decrease in LHRH release (regulated by nitric oxide) into blood vessels with an induced β-endorphin increase through stimulation of μ-opiate receptors [237]. Young rats subjected to 30-min exercise demonstrated an increased permeability of the blood–brain barrier to serotonin by 5-HT2 receptors following an increase in β-endorphin levels [238]. The variety in β-endorphin’s regulatory effects seem to place it near to many of the factors influencing neurodegenerative disorder.

#### 2.7.4. β-Endorphin-Based Treatment

A summary of some of the effects that β-endorphins may have on neurodegenerative disease is shown in Figure 2.

While it would seem that the overall impact of β-endorphins on the neurodegenerative process is beneficial, much of the research on opioid receptor agonists as a therapeutic target has been focused on inhibiting it, perhaps due to its significance in inflammatory action. Some research has shown that prolonged opioid abuse can exacerbate the physical hallmarks of AD, and postmortem analysis of opioid abusers’ brains show early signs of developing the disorder [239,240]. Administration of opioid system antagonists, such as naloxone and naltrexone, has been studied as a potential therapeutic agent in AD and dementia. A 1985 study investigating the use of naltrexone in ameliorating dementia symptoms found no significant difference in cognitive performance between those taking the opioid antagonist during a week long period and those given a placebo [241]. A second study in 1986 also found no cognitive benefit to the administration of opioid antagonist naloxone, though increases in behavioral arousal and psychomotor retardation at higher doses were noted [242]. Other studies examining the use of opioid antagonists find similar inefficacy [243,244]. While these studies did not investigate long-term β-endorphin inhibition, the application of opioid antagonists to benefit patients of neurodegenerative disease appears to be limited.

As with some psychiatric disorders, there is evidence that exercise may ameliorate neurodegenerative diseases such as AD, while a lack of physical activity can exacerbate one’s risk. AD patients are generally less physically active prior to their diagnosis, and longitudinal studies of cognitively normal adults found that regular exercise was associated with a reduction in the risk of developing AD, though not necessarily through β-endorphin-dependent mechanisms [245]. Exercise has also been found to promote the expression of various growth factors within the brain that counteract the neurodegenerative process of AD and general aging in the hippocampus, and promote neurogenesis [246]. As mentioned previously, exercise dependent neurogeneration was suggested to be dependent on β-endorphins by Koehl et al. [205]. Exercise also features a host of other benefits, such as decreases in Aβ levels and reduction in ROS generation [247], that make engaging in it regularly worthwhile from a neuroprotective perspective.

β-Endorphins are released in response to activation of the glucagon-like-peptide 1 (GLP-1) receptor [248]. GLP-1 is primarily released in response to meal ingestion; among its effects, the hormone notably stimulates the secretion of insulin and increases glucose sensitivity of pancreatic β-cells [249], GLP-1 receptors can be found within the brain. In rodent models, the activation of this receptor results in neuroprotective effects, including inhibition of glutamate-induced apoptosis [250]. Though a primary function of this release is to make use of β-endorphin’s antinociceptive properties, it has been shown that the neuroprotective effects of certain components released through GLP-1 activity, such as exenatide and catalpol, require β-endorphin presence to take effect, and are correspondingly neutralized by the administration of naloxone [251]. Recent studies have indicated that GLP-1 agonists may help to improve the condition of those with AD and Parkinson’s disease [252].

## 3. Concluding Remarks

β-Endorphins, and the system of opioid receptor agonists more generally, have a part to play in a wide variety of biological systems. While they are most well-known for their antinociceptive properties and stress-relieving nature, they also have their hand in homeostatic function and behavior. Their impact on the immune system is inhibitive overall, but situational and regional exceptions make further study imperative; though the mu-opioid receptor is its primary target, the peptide’s effects extend beyond it. Notable deviations in levels of β-endorphins found in sufferers of both psychological and neurodegenerative disorders provide evidence that the peptide may be a worthwhile therapeutic target. While they are the most common method of modifying β-endorphin activity, opioid antagonists, such as naloxone, have met with failure in producing significant improvement in patients with both psychiatric and neurodegenerative disorders, and little has been done to increase the peptide’s activity. There are considerable dangers in the use of exogenous opioids, and what evidence of their impact there is points to greater harm than benefit. More natural methods of modulating β-endorphin levels, such as exercise, have proven to be helpful in a variety of disorders.

## Figures and Tables

**Figure 1 ijms-22-00338-f001:**
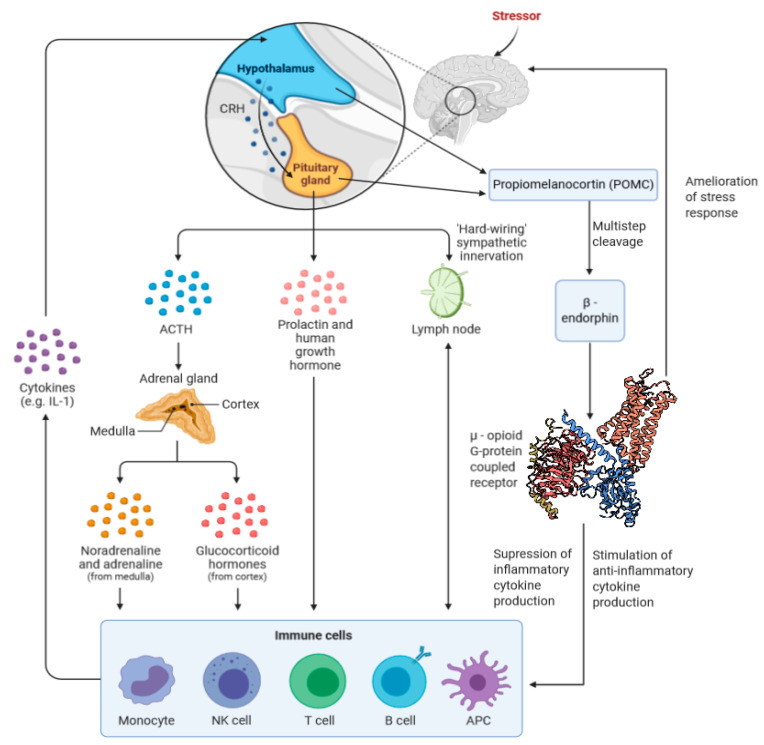
The interrelation between the hypothalamic–pituitary–adrenal (HPA), stress, and β-endorphins. β-Endorphins both attenuate the stress response and act to reduce levels of inflammatory cytokines, while increasing levels of anti-inflammatory cytokines, primarily through interaction with the mu-opioid receptor.

**Figure 2 ijms-22-00338-f002:**
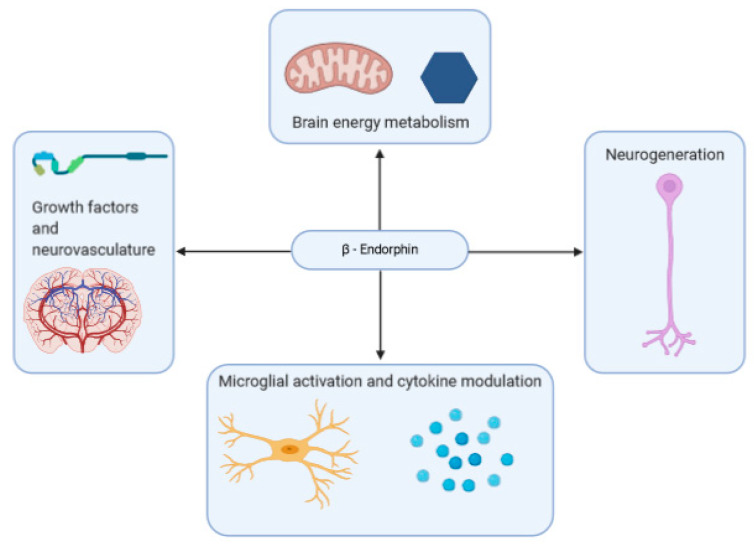
β-Endorphins exert a variety of effects on the brain that may impact brain health and neurodegenerative disorder.

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
