# Peer review of "Roles of β-Endorphin in Stress, Behavior, Neuroinflammation, and Brain Energy Metabolism"

_ijms, 2020, doi:10.3390/ijms22010338_

Round 1

Reviewer 1 Report

line 38 change 'have' to 'has'

Line 39, it would be useful to the reader name the other endogenous endorphins

Line 42, describes the HPA axis activation to stressful stimuli etc, but it does much more than this in terms of preparing the body for physiological change or adaptation to new environments, immune challenges etc. I think that the focus on 'stress' per se is a little disingenuous to the majority/range of HPA activities.

Line 45, name cleavage enzymes involved in formation of Beta endorphin and where they are located Anterior Pituitary, skin etc and the major source of brain derived Beta Endorphin as this dies not pass the BBB as far as I am aware

Section, 1.1 Demographics of β-endorphin levels, the authors should discuss whether changes the circulating plasma levels are a result of translation of the protein, expression of the cleavage enzyme or alterations in the degradation of the circulating beta endorphin protein with age.

Line 91, if they are going to include data from rodents and humans then the authors need to be clearer in each sentence whether they are describing results of human or none human studies.

Lines 120-126, is there more to add around exercise addiction/adherence and Beta endorphin levels?

Line 139, state origin of El-4 cell line, i.e. murine T cell phenotype etc

Line 144, change inhibitive to inhibitory

Lines 136-142, the authors should at least comment on the anti-inflammatory effects of Beta endorphin in the context of activation of the HPA axis and the release of the principal hormone cortisol

Line 170 change TFN to TNF

Line 177 Figure 1 legend, change to 'attenuate the stress response' rather than 'attenuate stress'

180 the brain. Benamar et. al. (state year)

Lines 179-195, is there any evidence that Beta endorphin can cross the BBB? In vitro experiments do not provide evidence of a valid physiological effect 

Line 197, effects (of) β-endorphins

Line 232, prolonged (or chronic) stressors.

Line 279, Variations or mutations?

Line 312, it is clear (that) interactions

Line 357 exercise was significantly more effective that (than) non-treatment

Line 358 performed in tandem -perhaps change to 'adjunct therapy'

Lines 362 to 365, there are 2 things happening here, participants getting used to exercise (a novel intervention or 'stress') may increase B end levels which may fall over time with regular exercise. You would need to look at a depressed group who exercise regularly compared to none depressed group who exercise regularly. I think that this fits well with your earlier comments around P End being released in acute stress events rather then chronic stress.

Lines 470-473, which fits well with a role in the acute stress response

Lines, 493, Although β-endorphin in its original form cannot be transported through the blood brain barrier due to its large size..... This point needs to be mentioned much earlier and the authors explain that the sources of B end secreted in the periphery do not directly bind to mu receptors in the brain. There are essentially 2 different stories here (1) peripheral B end and centrally released B end, the authors ned to discuss the sources of each. 

Given the point above the manuscript title is too narrow, given that it is peripheral sources of B End that are influencing the immune system and are being measured in plasma. A diagram showing the distinction between brain and peripheral sources of B End would make this clearer near the beginning of the manuscript.

Line 432, write in full 'Aβ levels' , not listed in abbreviations list

Line 535, I would argue that the principal role of GLP-1 is in glycaemic control rather then 'digestion' as the authors suggest. GLP-1 is a robust insulinotrophic peptide.

Line 566, all abbreviations should be listed in alphabetical order

Great diagrams throughout

Author Response

Reviewer 1’s comments and authors’ responses

line 38 change 'have' to 'has'

Response: We have corrected this error.

Line 39, it would be useful to the reader name the other endogenous endorphins

Response: We have added more information on the opioid receptor binding, and the receptors to which beta endorphin is known to bind.

Line 42, describes the HPA axis activation to stressful stimuli etc, but it does much more than this in terms of preparing the body for physiological change or adaptation to new environments, immune challenges etc. I think that the focus on 'stress' per se is a little disingenuous to the majority/range of HPA activities.

Response: We have made it more clear that the HPA is not only relevant to stress.

Line 45, name cleavage enzymes involved in formation of Beta endorphin and where they are located Anterior Pituitary, skin etc and the major source of brain derived Beta Endorphin as this dies not pass the BBB as far as I am aware

Response: Beta endorphin produced by the pituitary gland enters peripheral circulation. Beta endorphin is also produced in POMC neurons located primarily in the Hypothalamus, which we mention in the introduction. We have made it more clear where the respective sources of beta endorphin are located.

Section, 1.1 Demographics of β-endorphin levels, the authors should discuss whether changes the circulating plasma levels are a result of translation of the protein, expression of the cleavage enzyme or alterations in the degradation of the circulating beta endorphin protein with age.

Response: Ultimately studies relating to age looked primarily at levels of β-endorphin, and the cause cannot be extrapolated from that alone. We have added a brief comment on POMC expression, however.

Line 91, if they are going to include data from rodents and humans then the authors need to be clearer in each sentence whether they are describing results of human or none human studies.

Response: We have added labels indicating animal studies where they were missing.

Lines 120-126, is there more to add around exercise addiction/adherence and Beta endorphin levels?

Response: We have inserted a brief statement on this in the addiction section (2.4.1), but the idea has not been verified.

Line 139, state origin of El-4 cell line, i.e. murine T cell phenotype etc

Line 144, change inhibitive to inhibitory

Lines 136-142, the authors should at least comment on the anti-inflammatory effects of Beta endorphin in the context of activation of the HPA axis and the release of the principal hormone cortisol

Response: We have mentioned the release of cortisol alongside ACTH and the effect on inflammation.

Line 170 change TFN to TNF

Line 177 Figure 1 legend, change to 'attenuate the stress response' rather than 'attenuate stress'

180 the brain. Benamar et. al. (state year)

Response: We have made the requested changes.

Lines 179-195, is there any evidence that Beta endorphin can cross the BBB? In vitro experiments do not provide evidence of a valid physiological effect

Response: Beta endorphins secreted by the pituitary gland can make their way to the periphery, but radiolabeled beta endorphin injected into the carotid artery of mice can be extracted form brain tissue, indicating it can be transported across to a degree. We have added this information to the introduction.

Line 197, effects (of) β-endorphins

Response: We have reworded this section.

Line 232, prolonged (or chronic) stressors.

Response: We have corrected these errors.

Line 279, Variations or mutations?4

Response: We have changed this to “single nucleotide polymorphisms of … “

Line 312, it is clear (that) interactions

Line 357 exercise was significantly more effective that (than) non-treatment

Response: We have corrected these errors.

Line 358 performed in tandem -perhaps change to 'adjunct therapy'

Response: We have changed this to ‘combined with’.

Lines 362 to 365, there are 2 things happening here, participants getting used to exercise (a novel intervention or 'stress') may increase B end levels which may fall over time with regular exercise. You would need to look at a depressed group who exercise regularly compared to none depressed group who exercise regularly. I think that this fits well with your earlier comments around P End being released in acute stress events rather then chronic stress.

Lines 470-473, which fits well with a role in the acute stress response

Lines, 493, Although β-endorphin in its original form cannot be transported through the blood brain barrier due to its large size..... This point needs to be mentioned much earlier and the authors explain that the sources of B end secreted in the periphery do not directly bind to mu receptors in the brain. There are essentially 2 different stories here (1) peripheral B end and centrally released B end, the authors ned to discuss the sources of each. 

Given the point above the manuscript title is too narrow, given that it is peripheral sources of B End that are influencing the immune system and are being measured in plasma. A diagram showing the distinction between brain and peripheral sources of B End would make this clearer near the beginning of the manuscript.

Response: We have added more information regarding beta endorphin, the sources of it, and the blood brain barrier in the introduction. We have also altered the title

Line 432, write in full 'Aβ levels' , not listed in abbreviations list

Response: We have added Aβ to the abbreviations list.

Line 535, I would argue that the principal role of GLP-1 is in glycaemic control rather then 'digestion' as the authors suggest. GLP-1 is a robust insulinotrophic peptide.

Response: We agree that our indication of GLP-1’s function was inadequate and have changed it accordingly.

Line 566, all abbreviations should be listed in alphabetical order

Response: We have sorted the abbreviations list appropriately.

Great diagrams throughout

Response: Thanks for your nice comment.

Reviewer 2 Report

This review is on an important timely topic. However, there is a fundamental problem that requires a complete rewrite before this reviewer can consider it further. Specifically, throughout the review the authors refer to beta-endorphin as one molecule (although they occasionally use the plural "beta-endorphins" but never explain. Beta-endorphin is not a single molecule. It is essential that the authors describe the three major beta-endorphin peptides: full length 1-31, C-terminally truncated 1-29 and 1-28. In addition, nowhere in the review is any mention of the post-translational modification of N-terminal acetylation. In many brain regions, the major forms are C-terminally truncated 1-29 and N-terminally acetylated. These modifications have been reported to greatly affect the biological activity (although some of the old literature on 1-29 being an antagonist has not been found to be correct in several recent studies, and it is a full agonist).

These forms need to be introduced early in the review. Then, in subsequent sections, the authors need to state which specific form of beta-endorphin was studied (most likely the non-acetylated 1-31 form). But in many studies using RIAs, the various forms could not be distinguished, and in these cases this needs to be stated and then the generic 'beta-endorphins' is appropriate to use.

In addition to this major point, there's also a minor problem in the Introduction with the statement "β-endorphins are part of the system of endogenous opioids, a family that includes the endorphins, enkephalins, dynorphins, and endomorphins." This is not completely correct. Endomorphins are not likely to be endogenous opioids, as no precursor has been found despite decades of searching. A review should not repeat incorrect hypotheses, but should critically evaluate past claims. If endomorphins are mentioned at all (which is reasonable to include), the uncertainty over their endogenous status needs to be discussed.

Author Response

Reviewer 2’s comments and authors’ responses

This review is on an important timely topic. However, there is a fundamental problem that requires a complete rewrite before this reviewer can consider it further. Specifically, throughout the review the authors refer to beta-endorphin as one molecule (although they occasionally use the plural "beta-endorphins" but never explain. Beta-endorphin is not a single molecule. It is essential that the authors describe the three major beta-endorphin peptides: full length 1-31, C-terminally truncated 1-29 and 1-28. In addition, nowhere in the review is any mention of the post-translational modification of N-terminal acetylation. In many brain regions, the major forms are C-terminally truncated 1-29 and N-terminally acetylated. These modifications have been reported to greatly affect the biological activity (although some of the old literature on 1-29 being an antagonist has not been found to be correct in several recent studies, and it is a full agonist).

These forms need to be introduced early in the review. Then, in subsequent sections, the authors need to state which specific form of beta-endorphin was studied (most likely the non-acetylated 1-31 form). But in many studies using RIAs, the various forms could not be distinguished, and in these cases this needs to be stated and then the generic 'beta-endorphins' is appropriate to use. 

Response: We agree that the different forms of beta endorphin should have been elaborated in the introduction. However, after further review, the references/experiments cited here overwhelmingly do not specify any specific form, referring to the studied peptide as only “beta endorphin” or “beta endorphins”. As such, we cannot change the work to indicate the differences within each section beyond the introduction and analgesia components.

In addition to this major point, there's also a minor problem in the Introduction with the statement "β-endorphins are part of the system of endogenous opioids, a family that includes the endorphins, enkephalins, dynorphins, and endomorphins." This is not completely correct. Endomorphins are not likely to be endogenous opioids, as no precursor has been found despite decades of searching. A review should not repeat incorrect hypotheses, but should critically evaluate past claims. If endomorphins are mentioned at all (which is reasonable to include), the uncertainty over their endogenous status needs to be discussed.

Response: We have rewritten much of the introduction to beta endorphin, and have elaborated on the endomorphins as suggested.

Round 2

Reviewer 2 Report

This revised version is improved. The authors mention the diversity of beta-endorphin peptides. However, they are not up-to-date and cite some dogma that is no longer thought to be true. In my previous comments, I included the sentence "These modifications have been reported to greatly affect the biological activity (although some of the old literature on 1-29 being an antagonist has not been found to be correct in several recent studies, and it is a full agonist)." The authors ignored this last point about the old literature being questioned. They need to look this up and decide for themselves what to believe. Most importantly, they need to look at recent papers and also go back to the original papers that were behind the assumption that 1-29 is an antagonist. Remarkably, these old papers described weak studies done entirely in vivo, but somehow they became part of the dogma. Two recent papers have shown that 1-29 is a full opioid agonist. The most recent is Gomes et al, PNAS, 2020 "Biased signaling by endogenous opioid peptides". The other reference is cited within this paper. From the Abstract of the Gomes et al paper: "Our data also challenge the dogma that shorter forms of β-endorphin do not exhibit receptor activity; we show that they exhibit robust signaling in cultured cells and in an acute brain slice preparation."

Another myth perpetrated in the review is that the PCs convert precursors such as POMC into bioactive peptides. This is commonly stated in reviews on the PCs, but it is wrong. The PCs are ONLY the first step. There are other enzymes, such as carboxypeptidase E, and in many cases an amidating enzyme, that are needed to make the bioactive forms. It would be appropriate to mention these other enzymes in their Introduction.

Author Response

Reviewer 2’s comments and authors’ responses (2nd round)

This revised version is improved. The authors mention the diversity of beta-endorphin peptides. However, they are not up-to-date and cite some dogma that is no longer thought to be true. In my previous comments, I included the sentence "These modifications have been reported to greatly affect the biological activity (although some of the old literature on 1-29 being an antagonist has not been found to be correct in several recent studies, and it is a full agonist)." The authors ignored this last point about the old literature being questioned. They need to look this up and decide for themselves what to believe. Most importantly, they need to look at recent papers and also go back to the original papers that were behind the assumption that 1-29 is an antagonist. Remarkably, these old papers described weak studies done entirely in vivo, but somehow they became part of the dogma. Two recent papers have shown that 1-29 is a full opioid agonist. The most recent is Gomes et al, PNAS, 2020 "Biased signaling by endogenous opioid peptides". The other reference is cited within this paper. From the Abstract of the Gomes et al paper: "Our data also challenge the dogma that shorter forms of β-endorphin do not exhibit receptor activity; we show that they exhibit robust signaling in cultured cells and in an acute brain slice preparation."

Another myth perpetrated in the review is that the PCs convert precursors such as POMC into bioactive peptides. This is commonly stated in reviews on the PCs, but it is wrong. The PCs are ONLY the first step. There are other enzymes, such as carboxypeptidase E, and in many cases an amidating enzyme, that are needed to make the bioactive forms. It would be appropriate to mention these other enzymes in their Introduction.

Response: We have added information regarding the refutation of the shorter forms of β-endorphin acting as inhibitors, as well as more information on POMC processing, though we have made an effort to keep these sections brief.